# Potent Antiviral Activity of Vitamin B12 against Severe Acute Respiratory Syndrome Coronavirus 2, Middle East Respiratory Syndrome Coronavirus, and Human Coronavirus 229E

**DOI:** 10.3390/microorganisms11112777

**Published:** 2023-11-15

**Authors:** Yassmin Moatasim, Omnia Kutkat, Ahmed M. Osman, Mokhtar R. Gomaa, Faten Okda, Mohamed El Sayes, Mina Nabil Kamel, Mohamed Gaballah, Ahmed Mostafa, Rabeh El-Shesheny, Ghazi Kayali, Mohamed A. Ali, Ahmed Kandeil

**Affiliations:** 1Center of Scientific Excellence for Influenza Viruses, National Research Centre, Giza 12622, Egypt; yasmin.moatasim@human-link.org (Y.M.); omnia.abdelaziz@human-link.org (O.K.); mokhtar.rizk@human-link.org (M.R.G.); mohameddiaaelsayes@outlook.com (M.E.S.); mina@human-link.org (M.N.K.); gaballah09@gmail.com (M.G.); ahmed_nrc2000@hotmail.com (A.M.); rabeh.elshesheny@human-link.org (R.E.-S.); 2Biochemistry Department, Faculty of Science, Cairo University, Cairo 12613, Egypt; dr.ahmed.mamdouh.93@gmail.com; 3Veterinary Research Institute, National Research Centre, Giza 12622, Egypt; faten.okda@stjude.org; 4Department of Infectious Diseases, St. Jude Children’s Research Hospital, Memphis, TN 38105, USA; 5Human Link, Dubai 115738, United Arab Emirates; ghazi@human-link.org

**Keywords:** vitamins, antiviral agent, vitamin B12, SARS-CoV-2, MERS-CoV, viral infection

## Abstract

Repurposing vitamins as antiviral supporting agents is a rapid approach used to control emerging viral infections. Although there is considerable evidence supporting the use of vitamin supplementation in viral infections, including severe acute respiratory syndrome coronavirus 2 (SARS-CoV-2), the specific role of each vitamin in defending against coronaviruses remains unclear. Antiviral activities of available vitamins on the infectivity and replication of human coronaviruses, namely, SARS-CoV-2, Middle East respiratory syndrome coronavirus (MERS-CoV), and human coronavirus 229E (HCoV-229E), were investigated using in silico and in vitro studies. We identified potential broad-spectrum inhibitor effects of Hydroxocobalamin and Methylcobalamin against the three tested CoVs. Cyanocobalamin could selectively affect SARS-CoV-2 but not MERS-CoV and HCoV-229E. Methylcobalamin showed significantly higher inhibition values on SARS-CoV-2 compared with Hydroxocobalamin and Cyanocobalamin, while Hydroxocobalamin showed the highest potent antiviral activity against MERS-CoV and Cyanocobalamin against HCoV-229E. Furthermore, in silico studies were performed for these promising vitamins to investigate their interaction with SARS-CoV-2, MERS-CoV, and HCoV-229E viral-specific cell receptors (ACE2, DPP4, and hAPN protein, respectively) and viral proteins (S-RBD, 3CL pro, RdRp), suggesting that Hydroxocobalamin, Methylcobalamin, and Cyanocobalamin may have significant binding affinity to these proteins. These results show that Methylcobalamin may have potential benefits for coronavirus-infected patients.

## 1. Introduction

Severe acute respiratory syndrome coronavirus 2 (SARS-CoV-2) is the etiological agent for coronavirus disease 2019 (COVID-19). It was first detected in Wuhan, China, in late 2019, and soon after, it turned into a global pandemic causing nearly 673.68 million infections and more than 6,660,603 deaths to date (February 2023), as per the COVID-19 Data Repository by the Center for Systems Science and Engineering (CSSE) at Johns Hopkins University [1]. SARS-CoV-2 belongs to the Coronaviridae family of enveloped viruses, with a single-stranded positive-sense RNA genome [2,3]. Other members of the Coronaviridae family are also circulating worldwide in humans and causing illnesses that range from mild to severe. The Alphacoronavirus variants HCoVNL63 (Setracovirus) and HCoV-229E (Duvinacovirus) and Betacoronavirus variants HCoV-HKU1 and HCoV-OC43 (both in subgenus Embecovirus) cause mild respiratory diseases (e.g., the common cold). Other Betacoronavirus members, namely, SARS-CoV (sarbecovirus) and MERS-CoV (merbecovirus), cause more severe illnesses that lead to death in some cases [4,5].

Despite the continuous efforts to develop new and highly effective anti-SARS-CoV-2 drugs, potent antiviral agents against multiple coronaviruses remain elusive. Screening commercially available FDA-approved compounds remains the fastest way to find new anti-SARS-CoV-2 drugs [6]. The advantage of using approved drugs resides in their well-known clinical effects and toxicity profiles, which makes them easy to recruit for new purposes in health emergencies, especially if these compounds are commercially available and come at a low cost. The availability of repurposed drugs in all markets, especially in low-income countries, and their cost effectiveness remain major issues. Further, other factors include toxicity and uncertainty due to a lack of experimental validation.

Vitamins are organic molecules that are widely used as essential micronutrients that play an important role in cellular metabolism and have physiological effects on different biological responses, like host immunity. Thus, vitamin deficiency leads to an increased risk of developing infectious, allergic, and inflammatory diseases [7]. Vitamins are classified by their solubility into a fat-soluble group (vitamins A, D2, D3, and E) and a water-soluble group (vitamins B and C) [8]. Some vitamins were shown to have direct and indirect activity against many viral disorders. For example, Vitamin D can regulate immune functions by inducing the interferon signaling pathway and raising innate immunity against several viruses, such as HCV, rhinoviruses, and respiratory syncytial viruses [9,10,11,12]. In addition, Vitamin D also showed an anti-inflammatory response against many viral infections [11,12,13]. Vitamin D decreases the risk of many acute respiratory infections, and supplementation in healthcare workers reduces influenza-like illnesses [14,15]. In addition, Vitamin D has a direct antiviral activity against the hepatitis C virus [10,16].

Vitamins B1, C, D, and E play important roles in the innate immune system during viral infection [7,17,18]. Higher intakes of Vitamin C and Vitamin B complex micro-nutrients showed an impact on managing the COVID-19 pandemic successfully [19]. Vitamins B, E, and B9 increase the count number of circulating lymphocytic T cells, while Vitamin B6 is involved in lymphocyte proliferation. Vitamin A is required for immune cell maturation and functioning [20]. Vitamin A deficiencies are directly related to a higher severity of infections and impaired ability to regenerate the virus-damaged epithelia in children and animals [21]. Besides their role in boosting the antiviral immune response and controlling infection, vitamins have also been used in several treatment protocols [22].

Vitamin B12, which is sometimes referred to as cobalamin [9,23], is a water-soluble vitamin and has crucial roles in metabolic processes, the cardiovascular and circulatory systems, and the control of the immune system and antiviral activities [24]. Vitamin B12 is involved in repairing tissue damage and compensates for diminished hepatic storage during viral hepatitis.

Despite the huge amount of data that favors vitamin supplementation in the case of viral infections, and hence, SARS-CoV-2, the exact role of each vitamin in anti-coronaviruses is not clear. We aimed to study the effect of some vitamins on the infectivity and replication of human coronaviruses, namely, SARS-CoV-2, MERS-CoV, and HCoV-229E, using in silico and in vitro studies.

## 2. Materials and Methods

### 2.1. In Vitro Cytotoxicity and Inhibitory Activity against SARS-CoV-2 Virus

#### 2.1.1. Vitamins and Test Viruses

Raw reference vitamins used in the current study (*N* = 20) were provided by the National Organization for Drug Control and Research, Cairo, Egypt (Figure 1). All vitamin preparations were adjusted to 100 µM in 10% Dimethyl sulfoxide (DMSO) solution.

The CoV-19/Egypt/NRC-03/2020 SARS-CoV-2 (GISAID number: EPI_ISL_430819) [25], NRCE-HKU270 MERS-CoV (Genbank Accession: KJ477103.2) [26], and HCoV-229E were isolated and propagated in Vero E6 cells (ATCC CRL-1587) till the visualization of a cytopathic effect (CPE) [27]. Viruses were then titrated using TCID50 (50% tissue culture infectious dose) as previously described [28,29]. The HCoV-229E and Vero E6 cells were kindly provided by Dr. Leo Poon University of Hong Kong.

#### 2.1.2. Determination of In Vitro Cytotoxic Concentration 50% (CC50) and Antiviral Inhibitory Concentration 50% (IC50)

Cytotoxicity and antiviral activity of the tested vitamins were performed as previously described, with minor modifications [6,30]. Using 96-well tissue culture plates, 2.4 × 10^4^ Vero E6 cells were distributed in each well and incubated overnight in a humidified 37 °C incubator in a 5% CO_2_ atmosphere. After 24 h, cell monolayers were treated with a serial dilution of each vitamin and incubated for 72 h at 37 °C in parallel. Three sets of serial dilutions of each compound were prepared, and each was treated with 100 TCID50/well of the corresponding virus. After incubation, monolayers in 96-well plates were treated with the compound/virus mix and incubated for 3 days post-infection. Following incubation, the cells were fixed with 100 μL of 4% paraformaldehyde for 20 min. Cells were washed, and then they were stained with 0.1% crystal violet for 15 min at room temperature (RT). The crystal violet dye was dissolved using 100 μL of absolute methanol per well, and the optical density was measured at λ = 570 nm using the Anthos Zenyth 200rt plate reader (Anthos Labtec Instruments, Heerhugowaard, The Netherlands). The CC50 of the tested vitamins was the concentration required to reduce cell viability by 50%, while the IC50 of each vitamin was the concentration required to reduce the virus-induced cytopathic effect (CPE) by 50% relative to the virus control.

### 2.2. In Silico Analysis

#### 2.2.1. Ligand Preparation

The 2D structures of 20 ligands (Figure 1) were downloaded from the PubChem web server [31]. Then, Ligprep (Schrödinger Release 2020-3: LigPrep, Schrödinger, LLC, New York, NY, USA, 2020) was used to convert the ligands to 3D structures with possible conformations. A total of 569 conformers were generated. The compound IDs in PubChem are as follows: Alfacalcidol (5282181), Ascorbic acid (54670067), Biotin (171548), Cholecalciferol (5280795), Cyanocobalamin (5460135), Ergocalciferol (5280793), B9 (135398658), Hydroxocobalamin (70689311), methylcobalamin (10898559), Nicotinamide (936), Pantothenic acid (6613), Phylloquinone (5284607), Pyridoxal (1050), Retinol (445354), B2 (493570), Thiamine (1130), Tocopherol acetate (86472), Tocopherols (14986), vitamin K3 (4055), alpha lipoic acid (864), and benfotiamine (3032771).

#### 2.2.2. In Silico 3D Structure Protein Preparation

BioEdit (version 7.2.5) was used to align hCoV-19/Egypt/NRC-03/2020 of SARS-CoV-2 (GISAID ID: EPI_ISL_430819) and MERS-CoV (NCBI GenBank ID: KJ477103.2) with the reference genome (SARS-CoV-2 NCBI Reference Sequence: NC_045512.2 and MERS-CoV NCBI Reference Sequence: NC_019843.3) and with protein sequences obtained from the PDB bank protein of SARS-CoV-2 (surface protein ID: 6vsb, 3CL (3C-like protease) protein ID: 6y84 and RdRp (RNA-dependent RNA polymerase) protein ID: 6nur) and MERS-CoV sequences (surface RBD protein ID:4kqz, 3CL protein ID:5wkj) using Clustal Omega.

229E (surface protein ID: 6u7h and 3CL protein ID: 2zu2) and cell receptors (ACE2 angiotensin-converting enzyme 2 protein ID: 1r42, DPP4 Dipeptidyl peptidase-4 protein ID: 5y7k, and hAPN human aminopeptidase N protein ID: 5lhd) were used for docking.

Egyptian mutated SARS-CoV-2 S-protein, SARS-CoV-2 RdRp, MERS-CoV RBD, MERS-CoV 3CL, MERS-CoV RdRp, and 229E RdRp protein sequences obtained after alignment were sent for homology modeling.

Modeller 9.23 software [32] was used for the homology modeling of the 3D structure according to the Modeller basic modeling tutorial (https://salilab.org/modeller/tutorial/basic.html accessed on 10 October 2021). The SAVES webserver (https://saves.mbi.ucla.edu/ accessed on 10 October 2021) was used to verify the proteins generated as 3D structures using the Ramachandran plot prediction [33]. The model showed the highest number of residues in the favored and the allowed regions and the lowest number of residues in the outlier regions [34]. Preparation wizard software (Schrödinger Release 2020-3: Protein Preparation Wizard; Schrödinger, LLC, New York, NY, USA, 2020 version: 2020-3) was used for energy minimization, hydrogen addition, and water removal. Databases were accessed and data were retrieved and analysed at October 2021.

Active sites and binding pockets of spike proteins, polymerase, protease enzymes, and receptor proteins were selected and defined in the following publications [35,36,37,38,39,40,41,42,43,44].

#### 2.2.3. Molecular Docking

Glide software version 8.8—version number 88139 (Schrödinger Release 2020-3: Glide; Schrödinger, LLC, New York, NY, USA, 2020) was used for molecular docking procedures with a size of 20 Å around the active site, and other parameters for docking were set to the default value.

### 2.3. Quantitative Real-Time RT-PCR Assessment of RNA Expression after Treatment

Vitamins were prepared at 0.1 of CC50s in Dulbecco’s Modified Eagle Medium (DMEM) supplemented with 2% bovine serum albumin (BSA) and 1% Penicillin/Streptomycin antibiotic, mixed with the virus at a multiplicity of infection (MOI) of 0.05, and then incubated at RT for 1 h. In parallel, confluent monolayers of Vero E6 cells grown in 6-well plates were treated with 100 µL of the same concentrations of each compound for 1 h. This vitamin inoculum was then removed, and cells were infected with the prepared virus/vitamin mix and then incubated for 24 h. Vero-E6 infected cells without treatment were used as the untreated control. Supernatants of infected cells were collected and then subjected to viral RNA extraction using the QIAamp Viral-RNA Kit (Qiagen, Hilden, Germany). Virus RNA copy numbers were quantified with real-time qRT-PCR. The inhibition percentage of the viral RNA copy numbers was calculated as previously described [27] using the following formula:Percentage inhibition = (untreated − treated)/(untreated) × 100(1)

### 2.4. Antiviral Activity Using Plaque Reduction Assay

A plaque reduction assay was performed to confirm the antiviral activity of the three tested commercially available forms of Vitamin B12 as previously described [45]. Vero E6 cells were cultivated in 6-well plates till confluency. Viral dilution from each virus was prepared and then treated with concentrations of 10, 5, 2.5, and 1.25 µM of the tested compounds. The virus–vitamin mixes were then incubated for 1 h at 37 °C before being added to the Vero E6 cells. After 1 h of virus adsorption, the mixed inoculum was removed and fresh vitamin dilutions were added to their corresponding wells with the agarose/MEM overlay and the plates were left to solidify. Then, the plates were incubated at 37 °C for three days until the formation of viral plaques. The plates were fixed with formalin and visualized using crystal violet staining. Calculations were performed as follows:Viral inhibition percentage = (count of plaques in virus control − count of plaques in  virus with treatment)/(count of plaques in virus control) × 100

Plaque reduction assay was also performed for commercially available mixture of vitamins B1,B2, B3, B5 and B6 (B-Com, Amoun Pharmaceutical Industries Company, El Obour city, Egypt) against SARS-CoV-2 and MERS-CoV. Vitamin mix was purchased BN:225204, manufacturing date 10/2022. Molar concentrations of each vitamin in the mixture are illustrated in Appendix A.

### 2.5. In Vitro Mechanism of Action

Mechanisms of antiviral activity of vitamin B12 (represented by Methylcobalamin) were tested in dose-dependent viral inhibition assays on SARS-CoV-2, MERS-CoV, and HCoV-229E as previously described [46], with minor modifications.

#### 2.5.1. Viral Adsorption Mechanism

A triplicate of five dilutions of Methylcobalamin (25, 12.5, 6.25, 3.125, and 1.575 µM) was prepared and used to treat a confluent monolayer of Vero E6 cells in a 96-well tissue culture plate. Treated cells were incubated for 60 min at 37 °C; then washed three times with 1X PBS; and then infected with 100 µL of each virus, including 100 TCID50/well. Three wells of untreated cells were infected as the virus control, and three uninfected-untreated cells were used as the cell control. The plates were incubated for 60 min, then the inoculum was removed, and cells were supplemented with infection medium and incubated for 72 h. Cells were fixed with 4% formaldehyde solution, stained with 0.1% crystal violet, and quantified as previously described in the IC50 assay. The percentage inhibition was calculated as the reduction in treated cell death caused by the virus compared with the virus control.

#### 2.5.2. Viral Replication Mechanism

Confluent 96-well plates Vero E6 cells were infected with 100TCID50 of the viruses. After incubation for 1 h, the virus inoculum was removed and cells were then treated in triplicate with prepared Methylcobalamin concentrations (25, 12.5, 6.25, 3.125, and 1.575 µM) and incubated for 72 h, along with the virus control and cell control cells. Cells were fixed with 4% formaldehyde solution, stained with 0.1% crystal violet, and quantified as previously described.

#### 2.5.3. Neutralization Mechanism

A volume of 100 µL of 100TCID50/well of viruses was treated with Methylcobalamin concentrations (25, 12.5, 6.25, 3.125, and 1.575 μM) in triplicate in microtiter plate then incubated for 60 min at 37 °C. Methylcobalamin-treated viruses were inoculated in Vero E6 cells in confluent 96-well plates. After incubation at 37 °C for 60 min, the inoculum was removed and cells were washed with PBS, then supplemented with infection media, and incubated for 72 h. Cells were fixed with 4% formaldehyde solution, stained with 0.1% crystal violet, and quantified as previously described.

### 2.6. Statistical Analysis

GraphPad Prism 7 (GraphPad Software Inc., San Diego, CA, USA) was used to perform graphing and statistical analyses. Statistical analyses were performed using two-way ANOVA, followed by Bonferroni’s multiple comparisons test, where the confidence interval was set to 95%.

## 3. Results

### 3.1. Cytotoxicity and Inhibitory Activity of Reference Vitamins against Coronaviruses

Cytotoxicity in Vero E6 cells and antiviral activities of tested vitamins against the three CoVs were determined using a crystal violet assay. The results are shown in Table 1 and Figure 2 in terms of the half-maximal cytotoxic concentration (CC50), half-maximal inhibitory concentration (IC50), and selectivity safety index (SI = CC50/IC50).

Among the tested vitamins, tested forms of Vitamin B12 showed promising results against the tested CoVs. Hydroxocobalamin and Methylcobalamin showed significant activity against the three tested CoVs. On the other hand, Cyanocobalamin was active against SARS-CoV-2, with a selectivity index SI of 18.9, and mildly effective against the two other viruses, as shown in Table 1.

Vitamin B2 (Riboflavin Phosphate) showed high activity against HCoV-229E and SARS-CoV-2, with SI 57.5 and 22.7, respectively, and mild activity against MERS-CoV (SI 4.1). Vitamin B3 showed elevated safety index (SI) values against SARS-CoV-2 and HCoV-229E (39.2 and 29.5, respectively), while the Vitamin B5 SI values were elevated against MERS-CoV and HCoV-229E viruses (SI 42.0 and 14.6, respectively). Vitamins B1, D2, K3, and alpha lipoic acid showed strong antiviral activity against SARS-CoV-2 (SI 27.3, 13.5, 14.4, and 13.9, respectively). Alpha lipoic acid and vitamin E showed elevated anti-MERS activities (SI 16.2 and 17.5, respectively). Vitamin B6 showed antiviral activity against HCoV-229E (SI 54.5).

### 3.2. Titration of Viral RNA Inhibition Percentage Using qRT-PCR after Treatment

The results obtained from the RNA inhibition using RT-PCR showed that the highest inhibition (more than 70%) of SARS-CoV-2 replication could be achieved by treating with vitamins B1, B2, B3, B9, and B12 vitamins; and that of MERS-Co-V by treating with vitamins B2, B5, B6, B7, B9, B12, and E; and inhibition of HCoV-229E by treating with vitamins B2, B3, B9, and B12, as shown in Figure 3.

### 3.3. Protein Assembly and Preparation for Docking

Surface protein alignment of SARS-CoV-2 between the Egyptian sequence (GISAID ID: EPI_ISL_430819), SARS-CoV-2 NCBI reference sequence NC_045512.2, and surface protein from PDB bank (PDB ID: 6vsb) showed one amino acid mutation in the Egyptian spike surface protein (Spro) (D614G) and three amino acid mutation in 6vsb (R682G, R683S, and R685S). The alignment of the MERS-CoV surface proteins between the Egyptian sequence (NCBI GenBank ID: KJ477103.2), MERS-CoV NCBI reference sequence NC_019843.3, and RBD (PDB ID:4kqz) showed no mutation in the reference sequence and RBD from the PDB bank (PDB ID:4kqz) while having a single mutation in the Egyptian MERS-CoV RBD (A65T). RdRp and 3CL (Mpro) are considered to be conserved proteins within viruses [47,48]. However, after the alignment of the Egyptian sequences, the MERS-CoV NCBI reference sequence and MERS-CoV 3CL protein (PDB ID:5wkj) showed two amino acid mutations in 5wkj (I44M and S115A). The SARS-CoV-2 3CLs were 100% identical. The MERS-CoV RdRp was 100% identical to the Egyptian sequence and underwent homology modeling, given there was no presence of RdRp in the PDB bank databases. However, SARS-CoV-2 was slightly different, having nine mutations. All Ramachandran plots are displayed in Figure 4.

### 3.4. Molecular Docking

To determine the mechanism of action and the type of bonds, each vitamin was tested against the spike protein, 3CL (M-pro), the RdRP proteins of each virus, and the cell receptor specific to each type of tested coronavirus. The binding energies varied between different proteins. Figure 5 represents all binding energies as the Glide score. The docking of vitamin B12 (Hydroxycobalamin and methylcobalamin), B2, and B9 with the three virus proteins and the cell receptors for each virus are shown in Figure 6. The tables of binding energies and the type of binding of all tested compounds against all tested proteins are given in the Appendix A.

#### 3.4.1. Spike Proteins

For the SARS-CoV-2 S-protein, the binding energies (Glide score) ranged from −9.535 kcal/mol (hydroxocobalamin) to −1.405 kcal/mol (pantothenic acid). For the modeled SARS-CoV-2 S-protein, the binding energies ranged from −6.066 kcal/mol (B9) to no interaction (hydroxocobalamin, cyanocobalamin, and methylcobalamin). For the MERS-CoV S-protein, the binding energies ranged from −7.357 kcal/mol (hydroxocobalamin and cyanocobalamin) to −0.611 kcal/mol (pantothenic acid). For the modeled MERS-CoV S-protein, the binding energies ranged from −5.889 kcal/mol (B9) to −1.076 kcal/mol (pantothenic acid). For the 229E S-protein, the binding energies ranged from −10.678 kcal/mol (hydroxocobalamin and cyanocobalamin) to −1.657 kcal/mol (pantothenic acid). The types of bonds and binding energies are represented in Figure 6 and Figure 7.

#### 3.4.2. 3CL (M-Protease)

For the 3CL proteins, the SARS-CoV-2 binding energies ranged from −6.456 kcal/mol (B9) to no interaction (hydroxocobalamin, cyanocobalamin, and methylcobalamin). The MERS-CoV 3CL binding energies ranged from −10.704 kcal/mol (methylcobalamin) to −3.220 kcal/mol (pantothenic acid). The binding energies of the modeled MERS-CoV 3CL ranged from −10.267 kcal/mol (hydroxocobalamin and cyanocobalamin) to no interaction (retinol). The 229E 3CL binding energies ranged from −6.001 kcal/mol (Ergocalciferol) to no interaction (cyanocobalamin, hydroxocobalamin, methylcobalamin, vitamin k3, and phytomenadione). The types of bonds and binding energies are represented in Figure 6.

#### 3.4.3. Cell Receptors

The ACE2 receptor binding energies ranged between −9.589 kcal/mol (cyanocobalamin) and −2.572 kcal/mol (pantothenic acid). The DPP4 receptor binding energies ranged between −8.763 kcal/mol (cyanocobalamin) and −2.895 kcal/mol (phytomenadione). The hANP receptor binding energies ranged between −9.291 kcal/mol (methylcobalamin) and no interaction (vitamin k3). The types of bonds and binding energies are represented in Figure 6.

#### 3.4.4. RNA-Dependent RNA Polymerase (RdRP) Proteins

The SARS-CoV-2 RdRp protein binding energies ranged from −9.117 kcal/mol (hydroxocobalamin and cyanocobalamin) to −2.353 kcal/mol (pantothenic acid). The modeled SARS-CoV-2 RdRp protein binding energies ranged between −11.218 kcal/mol (hydroxocobalamin and cyanocobalamin) and no interaction (retinol). The modeled MERS-CoV RdRp protein binding energies ranged between −10.432 kcal/mol (hydroxocobalamin) and −1.956 kcal/mol (pantothenic acid). The modeled 229E RdRp protein binding energies ranged between −8.592 kcal/mol (hydroxocobalamin and cyanocobalamin) and −1.44 kcal/mol (pantothenic acid). The types of bonds and binding energies are represented in Figure 6.

### 3.5. Plaque Reduction Assay

The plaque reduction assay was also used to confirm the in vitro and in silico antiviral activities of the three tested forms of vitamin B12 against the three coronaviruses. The dose-dependent viral inhibition plaque reduction assay was used to compare the antiviral activity of the tested forms of vitamin B12 (Figure 7). The tested concentrations showed up to more than 80% inhibition against coronaviruses when compared with the untreated virus control.

Methylcobalamin showed significantly higher inhibition values on SARS-CoV-2 compared with the other forms of vitamin B12. Hydroxocobalamin had a significantly higher anti-MERS-CoV effect, and cyanocobalamin showed the highest antiviral activity against HCoV-229E.

### 3.6. Analyzing the Binding Capacity through Docking

#### 3.6.1. Spike RBD and Cell Receptors

Through docking studies, the SARS-CoV-2 spike protein and ACE-2 receptor were shown to form strong bindings with the three tested forms of vitamin B12 through H-bonds. Hydroxycobalamin was bound mainly to amino acids 496, 498, 503, and 505 of the spike and amino acids 31, 30, and 35 of the ACE2 binding site; cyanocobalamin was bound to amino acids 445, 489, and 446 of the spike and amino acids 30, 34, and 35 of the ACE2 binding site; and methylcobalamin was bound to amino acids 503, 505, and 493 of the spike and amino acids 30, 34, and 35 of the ACE2 binding site through hydrogen bonds. These amino acids are involved in the binding of the spike to the ACE-2 receptor [49]. Other amino acids are also involved, which means that vitamin B12 binds strongly to the RBD of the spike protein and to the receptor, which interferes with and blocks the binding of the RBD to the receptor. B2 binds to amino acids 496 and 503 of the spike RBD and amino acids 26, 30, and 33 of the ACE2 binding pocket through H-bonds. B9 binds to amino acids 505, 403, and 427 of the spike RBD and amino acids 30, 33, and 34 of the ACE2 binding pocket through H-bonds, hence interfering with the binding to the receptor pocket.

In the case of the 229E spike protein, the receptor binding domain is formed of three loops connecting six beta sheaths, loop 1 308:325, loop 2 352:359, and loop 3 404:408 [50]. Vitamin B12 forms binds through hydrogen bonds to residues 334, 389/390, 391/394/395, and 416 by concurrently binding to the three spike subunits in the trimer. This might interfere with the shape and charge of the formed trimer, and thus, the binding with the receptor. The same applies to B2 and B9. At the same time, the B12 vitamin hyroxycobalamin binds to residues 288, 289, and 291, and cyanocobalamin and methylcobalamin bind to 288, 291, 310, and 315. These residues are critical in the binding pocket of the hAPN and form Hydrogen bonds with loops 1 and 2 of the RBD [50], which interferes with the affinity and binding of the RBD to the receptor pocket. B9 and B2 form Hydrogen bonds to 289, 301, and 255 residues, which are involved in the structure of the pocket.

With regard to the MERS-CoV spike protein, residues 506, 553, 555, 540, 542, 510, 513, 511, 499, 539, 537, and 536 are critical in the binding and entry of the virus inside the cell by forming H-bonds, salt bridges, or hydrophobic interactions [40]. Methylcobalamin binds to the receptor binding domain through residues 499, 501, 540, 537, and 542 using H-bonds, and to amino acid 553 through hydrophobic interactions. This hinders the binding and entry of the virus by blocking the binding site in the receptor binding domain. Both hydroxycobalamin and cyanocobalamin bind to residues 540, 542, 511, and 466 using hydrogen bonds, thus interfering with the ability of RBD to bind to the receptor. B2 binds to site 510 and interferes with the formation of salt bridges with amino acid 317 of the DPP4 receptor and hinders virus binding and entry. B9 binds to residues 536 and 538, hence interfering with the salt bridge formation with the 267 residue of DPP4, and binds to amino acid 540 in RBD, which is responsible for stabilizing the hydrophobic core that is critical for DPP4 receptor binding and viral entry in the RBD. Residues 506, 553, and 555 in the RBD form hydrophobic interactions with residues 294 and 295 of the DPP4, forming the hydrophobic core stabilized by hydrophilic residues of RBD 510, 513, and 540 and residues 298, 317, 344 of the receptor [40]. The three forms of vitamin B12, B2, and B9 also bind to the receptor pocket with five to six hydrogen bonds, hence interfering with virus binding and entry. Hydroxycobalamin binds to residue 294, which forms a hydrophobic interaction with the RBD of SARS-CoV-2 [40]. Cyanocobalamin binds to residues 317 and 344, thus blocking the salt bridge that is critical for binding with RBD in residues 510 and 513 [38,40]. B9 also binds to residue 344.

#### 3.6.2. RNA-Dependent RNA Polymerase

Through docking analysis, hydroxocobalamin was shown to interact with some catalytic residues in the active pocket of SARS-CoV-2 RdRP (618, 622, 760, 811, 814, and 553) via H-bonds, thus blocking the access of RNA to the active pocket and interfering with the formation of a complex with NSP7/NSP8 [38]. While cyanocobalamin binds to residues 551, 618, and 760, methylcobalamin binds to residues 621, 811, 623, and 555 in the active site through H-bonding.

In the case of MERS-CoV RdRP, the three B12 vitamins were found to form several Hydrogen bonds with the active site, thus interfering with the binding of active site residues to the Mg^2+^ cofactor and also binding with the ligand [39]. Hydroxocobalamin binds to residues 306, 307, 113, 117, 255, and 48, cyanocobalamin binds to residues 86, 88, 255, and 760 and methylcobalamin binds to residues 113, 306, and 46, thus interfering with the active site. B2 and B9 bind to the conserved active aspartates D255 and D256, and interfere with the proper folding of the surface loop and with binding as H-bonding interactions or the metal (two Mg^2+^) interaction [39]. They were also found to form several hydrogen bonds with the active site and ensure stable binding.

#### 3.6.3. 3CL Protease

The three tested forms of vitamin B12 bound effectively only to the active site of 229E-3CLpro with the catalytic dyad at Cys145, forming a H-bond.

### 3.7. In Vitro Mechanism of Action

The results of screening for vitamins as direct antiviral agents, IC50, viral copy numbers reduction, and protein docking studies (on viral proteins and viral receptors on VERO E6 cells) suggest that vitamin B12 had direct antiviral activity against SARS-CoV-2, MERS-CoV, and HCoV-229E. To confirm the results obtained from these studies, three modes of action were tested in dose-dependent viral inhibition assays. The results reveal that B12 attenuated the coronaviruses via three different modes of action in a dose-dependent manner (Figure 8). Viral infectivity was inhibited by B12 molecules binding to spike RBD proteins (neutralization) and cell receptors (adsorption block) that partially broke down the spike RBD affinity to cellular receptors and vice versa. B12 also interfered with the replication steps within the infected cells, as they interacted with 3CL pro and RdRp polymerases.

## 4. Discussion

Vitamins are vital for cell growth, function, and development. Furthermore, they play an important role in responses to pathogens via cell-mediated responses and boosting immunity. Almost all vitamins have been reported to increase antibody production. Many also have key roles in innate and adaptive immunity. In addition, vitamin C is an antioxidant, anti-inflammatory, anti-coagulant, and has immunomodulatory functions. Vitamins reduce the risk of pneumonia and other viral respiratory tract infections [51]. They can be a potential source for the development of antiviral therapeutics against SARS-CoV-2 alone or when combined with other antiviral drugs against coronaviruses. The role of food supplements and vitamins in the prevention and treatment of respiratory tract infections in the general population was discussed in several studies during the COVID-19 pandemic [52].

In the current study, the direct antiviral activity of some vitamins against several coronaviruses was assessed, both in vitro and in silico. The current study showed that some vitamins had a direct effect on viral infectivity and replication, and they downregulated the viral RNA of SARS-CoV-2, MERS-CoV, and HCoV-229E in infected Vero E6 cells. These direct antiviral activities tended to be dose-dependent in vitro to inhibit viral infection and replication. Our results indicate only the direct antiviral activity, as some of the tested vitamins might show low or no antiviral activity but still contribute to patient recovery indirectly by boosting the immune response. In our study, vitamin C showed low direct anti-coronavirus activity and high doses were required. The results of some clinical studies of high-dosage vitamin C in COVID-19 patients showed nonsignificant differences in therapy durations [53,54]. However, vitamin C was shown to be linked to lower mortality and severity by boosting immune antiviral activity in SARS-CoV-2 patients [55,56].

Vitamin B12 (also known as cobalamin) is readily available from multiple sources, it is affordable, can be self-administered by patients, is available worldwide, and displays low-to-no toxicity at high doses [57]. The Methylcobalamin form is the metabolically active form, while hydroxycobalamin and cyanocobalamin must be converted to methylcobalamin or 5-deoxyadenosylcobalamin to become active [57]. Cyanocobalamin is a synthetic form of vitamin B12 that contains a cyano group, while other forms can occur naturally. Generally, Vitamin B12 is considered an immunomodulator, especially with respect to CD8+ cells and the NK cell system [7]. Vitamin B12 has been used as a supplement against many viral infections (such as hepatitis, HIV, and Norovirus) [19]. It was reported to play an important role against SARS-CoV-2; however, the mechanism has not been elucidated yet. Molecular modeling showed the potential use of Vitamin B12 in the treatment of COVID-19 patients [58].

Some SARS-CoV-2 proteins, such as ORF3a and ORF7a, are linked to elevated NFκB levels and induction of the proinflammatory immune response, which is linked to lung inflammation, severity, and ventilation in COVID-19 patients [59,60,61]. In the case of SARS-CoV, controlling the NFκB expression reduces lung inflammation and lowers the severity of infection in mice [62]. The use of vitamin B12 is linked to regulating NFκB levels affecting the expression of genes encoding pro-inflammatory cytokines, and in clinical trials, Vitamin B12 therapy decreased the uses of mechanical ventilatory support [17,63,64]. Patients with COVID-19 and post-COVID-19 exhibit symptoms that are similar to those found in vitamin B12 deficiency [19,65,66]. B12 was shown to relieve symptoms that could also be related to a COVID-19 prognosis, such as decreased pain intensity as an analgesic [67], lowered depressive symptoms [68,69], decreased memory loss, and less impaired concentration [19]. Vitamin B12 therapy reduces oxidative stress, improves circulation, and acts as an anti-inflammatory and analgesic, thus probably reducing the damage to infected cells [70].

In this study, in silico screening results of vitamins’ interactions with SARS-CoV-2, MERS-CoV, and HCoV-229E viral-specific cell receptors (ACE2, DPP4, and hAPN protein respectively) and viral proteins (S-RBD, 3CL pro, RdRp) suggest that vitamin B12 in three different tested forms may have significant binding affinities to these proteins, which explains the adsorption inhibition activity by blocking viral specific cell receptors and neutralization activities by binding to S-RBD active sites. The notable inhibition of viral RNA copy numbers and viral replication inhibition rates with the three forms of B12 may have been due to the significant binding affinities of the three compounds to the viral RdRp polymerase. Pandya, Shah et al. showed that B12 also exhibits strong binding to furin endoprotease protein, which could stop the cleavage of the spike protein and may be the component that reduces virulence [7].

Methylcobalamin showed significant affinity to the active site of the nsp12 protein of SARS-CoV-2 in this study and previous work from Pandya et al. [7]. While our in silico analysis showed no binding to the 3CL protease, except for MERS-CoV, other variants of SARS-CoV-2 showed strong binding to the active site of NSP5 [7]. Our in vitro findings are in agreement with the computational analyses that showed that tested forms of vitamin B12 resulted in significant binding. The in vitro mode of action of Methylcobalamin confirmed the viral inhibition at various stages. Our data reveal that B12 vitamins are efficient against most coronavirus protein targets.

Our data shows that other vitamins (B2 and B9) also showed significant inhibition of various proteins in the three coronaviruses, both in vitro and in silico. B9 is known to increase the activity of natural killer cells and raise the count of circulating lymphocytic T cells, constituting significant antiviral activity [7,18]. Another in silico analysis showed that B9 binds effectively to the furin endoprotease, which is essential for SARS-CoV-2 infection and virulence [7]. Our data support the wide and multi-protein target antiviral activity of vitamin B9.

B2 is important in many cellular functions, cell growth, and energy metabolism, and is involved in the metabolism of various vitamins [71,72]. It is also essential as an anti-oxidative agent, for the activation and enhancement of macrophage functions, the suppression of pro-inflammatory cytokines, and increases host resistance toward microbial infections. The photoactivated B2 was shown to have antimicrobial activity, and when combined with UV in human plasma products, it reduced the replication in MERS-CoV and SARS-CoV-2 to a non-detectable level [7,71,73]. A clinical study showed decreased inflammation and normalization of neutrophil and lymphocyte counts, as well as their relative ratio in COVID-19 patients when soluble B2 was administrated [74]. Here, we showed the direct antiviral activity of vitamin B2 via inhibition of the RNA copy number in vitro, and through in silico studies by binding efficiently to the active sites of RdRP and 3CL protease, and by interfering with the spike RBD, cell receptor affinity, and efficiency of binding [18].

The anti-coronavirus activity of a commercial form of an injectable vitamin b complex (B-Com by Amoun pharm), which was composed of B1, B2, B3, B5, B6, showed effecient inhibition of viral replication using a plaque assay of SARS-CoV-2 and MERS-CoV (Appendix A), which favors the combined use of a B complex for protection or treatment [75].

Vitamin C showed low in vitro inhibition of virus replication and low binding affinity to the proteins of the three coronavirus proteins in silico. In clinical trials, vitamin C was shown to have no activity in treatment or protection against the common cold [76] or against SARS-CoV-2 [54], but high doses were linked to a reduced risk of 28-day mortality [77].

The data from clinical trials were not consistent regarding vitamin D3. Many studies linked the severity of infection with low plasma levels of D3, and some recommended its use as a prophylaxis [78]. However, many clinical studies showed that D3 supplementation led to a significant lowering of COVID-19-related events [79]. In many studies, D3 reduced the rate of ICU administration or hospital stay, but not significantly [80,81,82,83,84]. In this study, in vitro analysis showed moderate inactivation of the virus replication and low binding affinity to viral proteins in silico. This could mean that antiviral D3 activity might be indirect through an innate immune response.

There is an urgent need for the identification of new molecules that can reduce viral titers, and thus, limit the severity of new emerging viral diseases. Importantly, we found that different forms of vitamin B12, namely, Cyanocobalamin, methylcobalamin, and hydroxocobalamin, which are used orally and intravenously worldwide, inhibited the replication of SARS-CoV-2, MERS-CoV, and 229E-HCV in vitro. Our results indicate that many other vitamins, such as vitamins B2, B3, B5, and B9, also showed effective direct antiviral activity against some coronaviruses, which could suggest the possibility of higher antiviral activity in a combination of different B vitamins. However, more in vivo studies are needed to investigate the appropriate choice of the chemical form of vitamin B12, dose, and supplementation time, or whether it should be used alone or in combination with other vitamins. Further examinations in this field are needed, and the new information uncovered will help to develop a new era of precision health and nutrition.

## Figures and Tables

**Figure 1 microorganisms-11-02777-f001:**
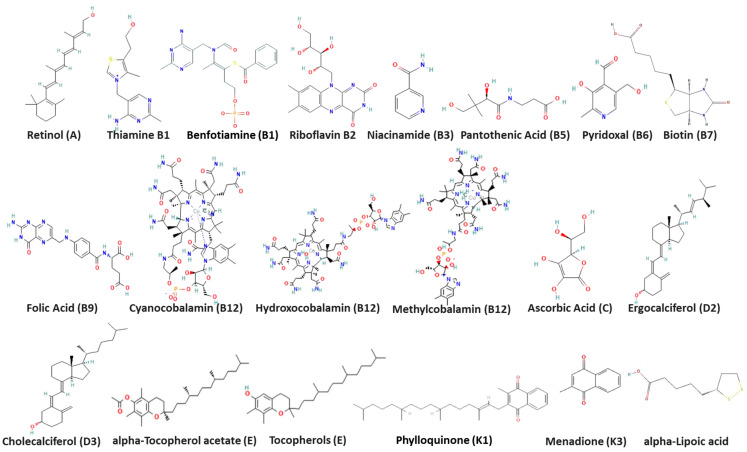
Two-dimensional chemical structures of vitamins included in this study as obtained from PubChem web server.

**Figure 2 microorganisms-11-02777-f002:**
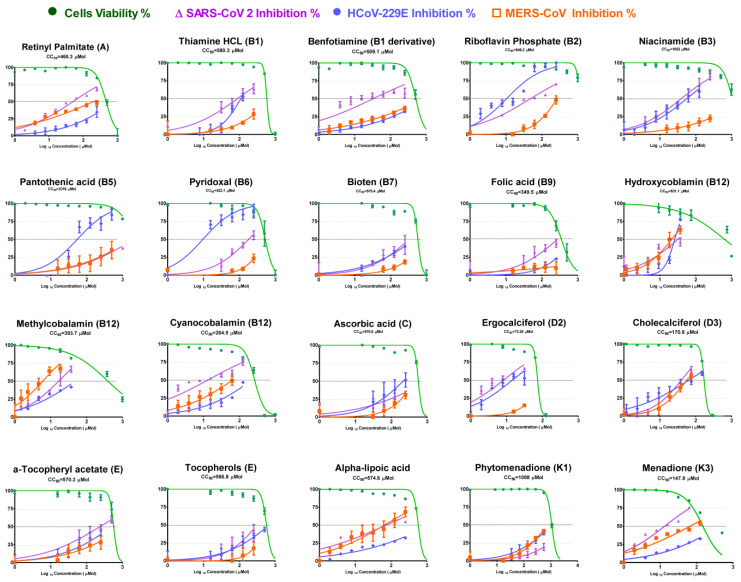
Determination of CC50 and IC50 of vitamins in Vero E6 cells against SARS-CoV-2, MERS-CoV, and HCoV-229E. Values of CC50 and IC50 were calculated using non-linear regression analysis with Graph Pad Prism software (version 5.01) by plotting log inhibitor versus normalized response (variable slope); All values are listed in Table 1.

**Figure 3 microorganisms-11-02777-f003:**
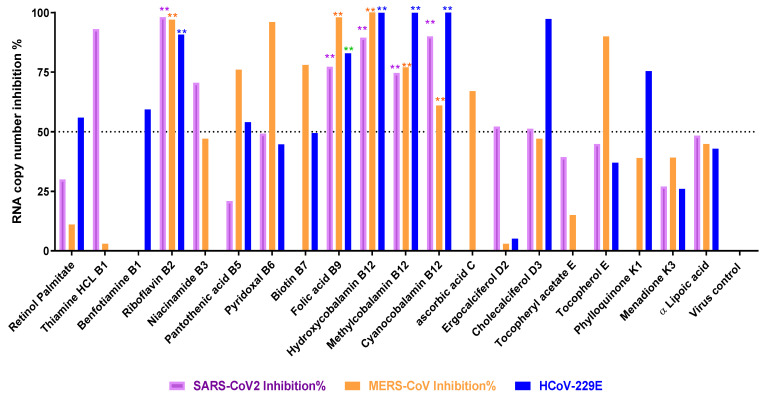
Viral inhibition percentage of viral RNA copy numbers of SARS-CoV-2, MERS-CoV, and HCoV-229E at 24 h post-infection and treatment with vitamins using Vero E6 cells (MOI 0.05). Real-time RT-PCR was used to compare the treatment with untreated virus control. Small stars represent the level of significance compared with the virus control tested using GraphPad prism software v7 (2-way ANOVA test with Dunnett’s post-test and 95% confidence level). Colors of stars match the color of specific column. Significance values are represented as follows, ** refers to *p* ≤ 0.01.

**Figure 4 microorganisms-11-02777-f004:**
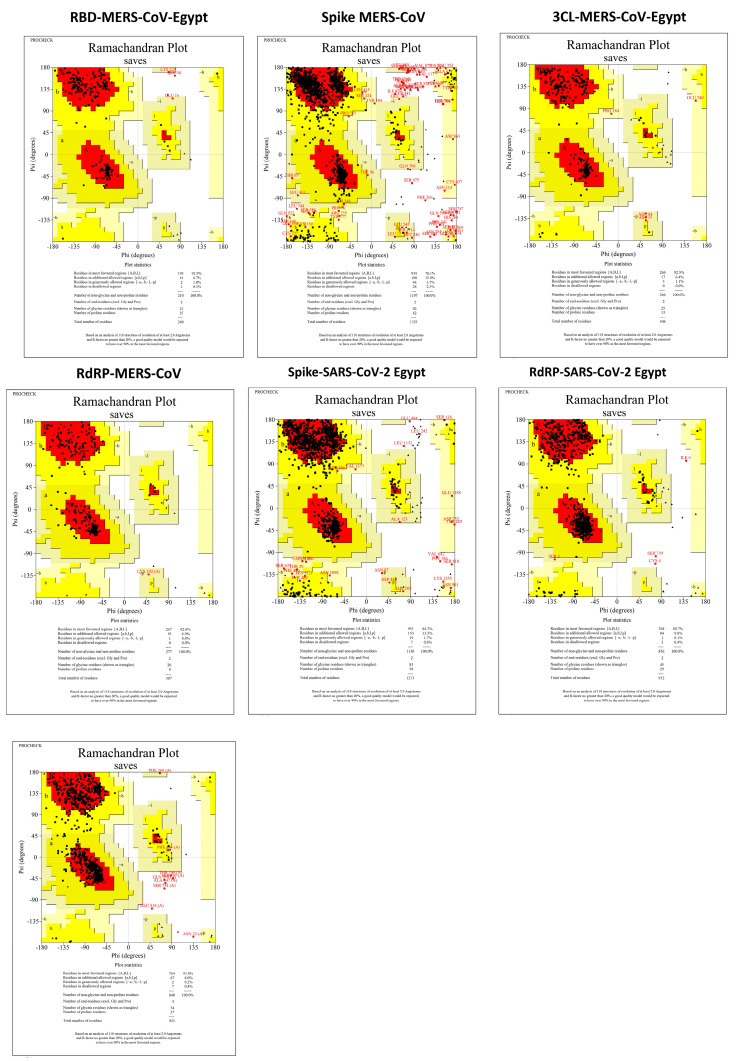
Ramachandran plot showing the phi/psi values of the modeled proteins of the three coronaviruses. Based on analyses of 118 structures with resolution of at least 2.0 angstroms and R—factor no greater than 20%, a good quality model was expected to have over 90% in the most favored regions. Red colorfavorable regions are represented by red color, while the yellow color is the allowed region, and white color is the disallowed regions. Black dots are the amino acid residues.

**Figure 5 microorganisms-11-02777-f005:**
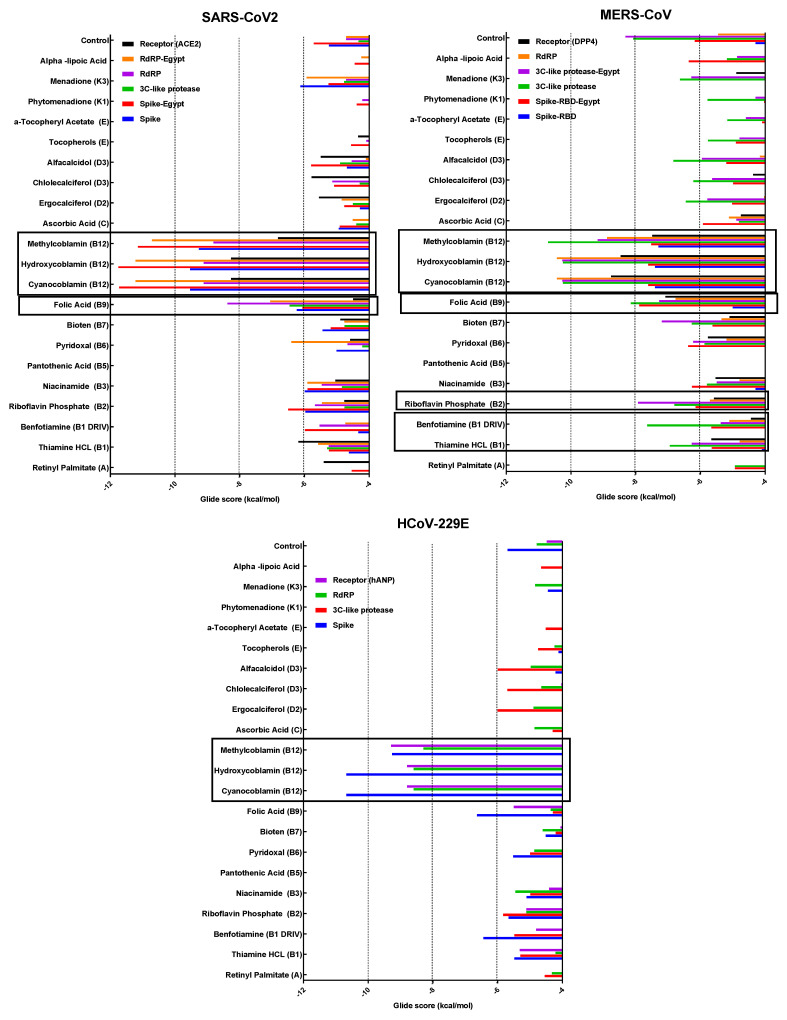
Binding energies (as a Glide score) of the tested vitamins against different proteins correlated with the three coronaviruses. All score values are in negative charge. The space between the bars in each vitamin means no binding was detected or binding energies were higher than −4 kcal/mol (less stable binding). Black boxes indicate the lowest, i.e., strongest, binding. Egypt in protein names indicates the 3D modeling protein from sequences of isolated MERS and SARS-2 viruses from Egypt involved in this study in in vitro experiments, while other proteins were retrieved as 3D structures from the PDB as reference strains.

**Figure 6 microorganisms-11-02777-f006:**
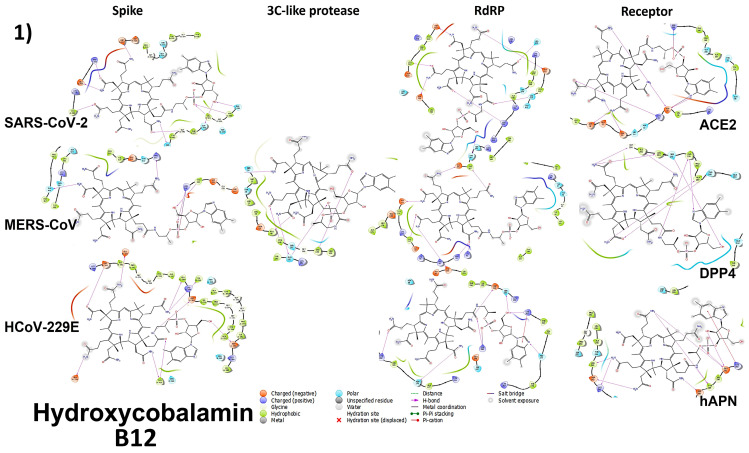
Molecular docking of the 3 forms of B12 vitamin (**1**–**3**), B2 (**4**), and B9 (**5**) on the viral proteins and cell receptors of SARS-CoV-2, MERS-CoV, and HCoV-229E. Horizontal axes represent the viruses and the vertical axes represent the proteins. Empty spots represent no binding. Purple lines represent the formation of Hydrogen bonds. Red and blue residues are charged ones, light blue are polar ones, and green ones are hydrophobic.

**Figure 7 microorganisms-11-02777-f007:**
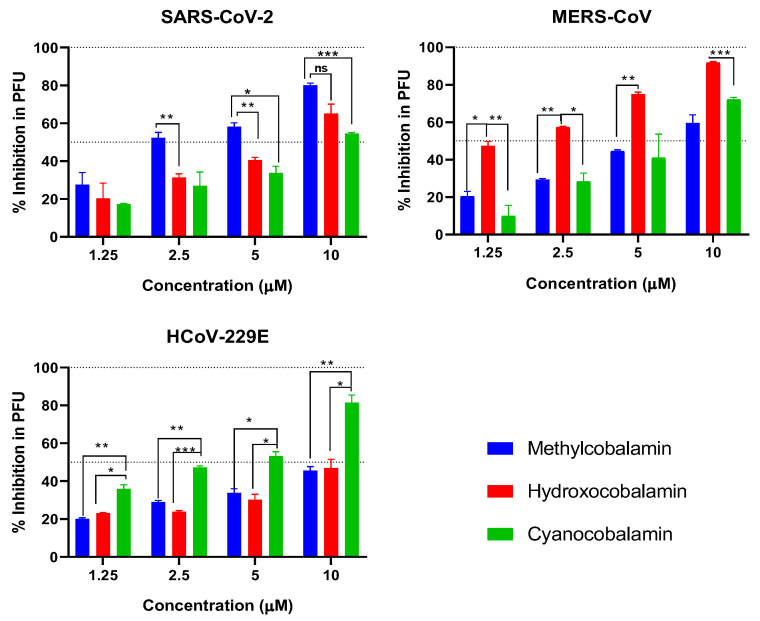
Plaque reduction assay of three tested forms of vitamin B12, namely, Methylcobalamin (blue), hydroxocobalamin (red), and cyanocobalamin (green), against SARS-CoV-2, MERS-CoV, and 229E viruses at doses of 1.25-2.5-5-10 micromoles. Statistical analyses were performed using two-way ANOVA, followed by Bonferroni’s multiple comparisons test, where the confidence interval was set to 95%. * refers to *p* ≤ 0.05, ** refers to *p* ≤ 0.01, *** refers to *p* ≤ 0.001.

**Figure 8 microorganisms-11-02777-f008:**
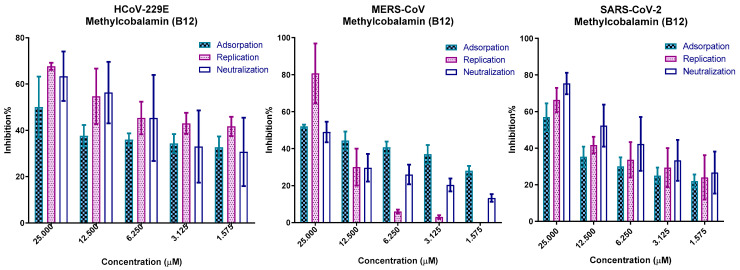
Mode of action of Methylcobalamin against the three coronaviruses on VERO-E6 cells, showing that vitamin b12 reduced viral replication in different stages in viral replication cycle of the three tested coronaviruses.

**Table 1 microorganisms-11-02777-t001:** The half-maximal cytotoxic concentration (CC50) in µM, half-maximal inhibitory concentration (IC50) in µM, and selectivity safety index (SI) values of the tested vitamins against SARS-CoV-2, HCoV-229E, and MERS-CoV. SI values less than or equal to 1 indicate no antiviral activity, while higher SI values indicate higher efficiency.

Chemical Name	Generic Name	CC50	SARS-CoV-2		MERS-CoV		HCoV-229E	
			IC50	SI	IC50	SI	IC50	SI
Retinyl Palmitate	A	466.3	41.0	11.4	125.0	3.7	345.9	1.3
Thiamine HCL	B1	580.3	52.0	11.2	128.7	4.5	79.5	7.3
Benfotiamine (Thiamine)	B1	509.1	18.6	27.3	387.1	1.3	525.5	1.0
Riboflavin Phosphate	B2	848.2	37.4	22.7	208.0	4.1	14.8	57.5
Niacinamide (niacin)	B3	1052.0	26.9	39.2	219.7	4.8	35.6	29.5
Pantothenic Acid	B5	2316.0	917.8	2.5	55.1	42.0	158.3	14.6
Pyridoxal	B6	523.1	110.6	4.7	267.2	2.0	9.6	54.5
Biotin	B7	575.4	129.1	4.5	317.3	1.8	>CC50	0
Folic acid	B9	349.5	110.5	3.2	>CC50	0	330.6	1.1
Hydroxycobalamin	B12	631.1	17.0	37.1	21.5	29.3	26.1	24.2
Methylcobalamin	B12	393.7	12.0	32.9	6.0	65.4	29.7	13.3
Cyanocobalamin	B12	246.9	13.1	18.9	29.6	8.3	104.2	2.4
Ascorbic Acid	C	570.5	168.4	3.4	165.4	3.4	81.8	7.0
Ergocalciferol	D2	73.4	5.4	13.5	14.6	5.0	42.0	1.7
Chlolecalciferol	D3	170.8	29.8	5.7	38.6	4.4	33.9	5.0
a-Tocopheryl Acetate	E	570.2	122.3	4.7	104.5	5.5	127.9	4.5
Tocopherols	E	566.8	106.0	5.3	36.0	15.7	286.4	2.0
Alpha lipoic Acid	a-lipoic Acid	574.6	41.2	13.9	35.4	16.2	288.3	2.0
Phytomenadione	K1	1088.0	297.3	3.7	474.9	2.3	473.7	2.3
Menadione	K3	147.8	10.3	14.4	45.2	3.3	156.8	0.9

## Data Availability

All data is available in the manuscript.

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
