# Peer review of "Potent Antiviral Activity of Vitamin B12 against Severe Acute Respiratory Syndrome Coronavirus 2, Middle East Respiratory Syndrome Coronavirus, and Human Coronavirus 229E"

_microorganisms, 2023, doi:10.3390/microorganisms11112777_

Round 1

Reviewer 1 Report

Comments and Suggestions for Authors

This extraordinary study set out to examine whether commonly used B vitamins could inhibit three major coronaviruses of humans – SARS-CoV2, MERS-CoV and HCoV-229E – in vitro.  Surprisingly, they discovered that indeed, a number of B vitamins did inhibit these viruses and they demonstrated this by plaque reduction assay.  The scientists then set out to establish, via computer model, whether the possible virus inhibition was by simple binding and if so – binding to what?  They found that the three versions of vitamin B12 commonly available as supplements bound to the virus spike protein, host cell receptors, and RdRP enzyme, but not the 3C-like protease, and this was beautifully illustrated in Figure 6.  In their models vitamins B2 and B9 did bind to the 3C-like protease in addition to the others listed in the previous sentence. 

I had written in this review that this paper is an amazing work, very well written, and a valuable addition to the coronavirus literature because there are very few papers on this subject.  However, when I got to the Discussion, I discovered that many references were mis-represented, were misleading, or were even downright lies, which has caused me to question the integrity of the authors and the rest of the paper.  An example is that reference 58 did not include SARS-CoV 2: it is a study which finished in December 2015, well before SARS-CoV2.  This is very disappointing.

The ordering of your paper confused me a little: did you first do the computer models to determine which vitamins were worth testing by plaque assay, or did you first find out that some vitamins inhibit coronaviruses, then set out to determine how?  The order in Methods suggests the latter, but the order in Results suggests the former.  Please re-arrange the paper to conform to what actually happened.

The results of this study suggest that B12 supplement manufacturers would be best to offer a combination of different B12 compounds, not just a single one, if anti- coronavirus activity were their goal (of course B12 is usually taken to aid red blood cell production and nerve health).

I found myself wondering why the authors didn’t also test vitamin C because that is the classic supplement used to prevent and treat colds and flu-like diseases: perhaps you have published that already?  Please explain the reason.  Likewise why not test vitamin D which is known to have prevented COVID in people who had healthy levels?

You wisely considered that the vitamins might be interfering with the viral polymerase enzyme, rather than or in addition to, preventing attachment to cellular receptors, but did you not consider that you could have done a quantitative RT-PCR for messenger RNA on the cell culture supernatant to definitively prove that the vitamins interfered with viral replication?

Why did you dissolve the vitamins in DMSO when they are fat or water soluble?  Did you also control for DMSO cytotoxicity?

Did none of the vitamins form disulphide bonds with the virus or receptors? 

The paper tends to say “the three vitamin B12s” as if there are only three versions, please make it clear that you are referring simply to the three you tested, for example in Line 406 you’d say something like “The three [not 3] forms of vitamin B12 we tested, riboflavin, etc.”  Please include in the introduction an explanation of the differences between the various commercially available forms of B12: e.g. whether or not they are methylated or are coenzyme forms.

You tend to jump back and forwards between using the B number and the full name, which is inconsistent and only necessary for the B12 vitamin sub-types. Be consistent: e.g. keep in vitro and in vivo in italics throughout.

Throughout the paper, please check that you have explained abbreviations on first mention, e.g. ACE2 and DPP4 receptors. 

Line 51.  This isn’t quite true, is it?  Molnupiravir and GS-441524 seem to be promising antivirals against coronaviruses.

Line 65: some vitamins missing from your list, e.g. K2.  Are all the B vitamins water soluble?  I’m not clear why you mention vitamins that were not tested in your study.

Line 71: did you intend to omit from this paragraph the fact that normal vitamin D levels prevented COVID?  I am aware you may have done so for political reasons.  I know it’s vaguely covered by your “ influenza like illnesses statement”, but I would like to see you cite some of the SARS-CoV2 papers on it.

Line 73: why put ascorbic acid after C, but not describe B1?

Line 106: don’t you mean 10 to the power of 4?  i.e. you need to make the 4 superscript.

Line 108: it’s normal to make the 2 subscript.

Line 125: specify the antibiotic used because some antibiotics have anti-coronavirus activity.

Figure 1 – do you need a reference for these figures or did you make them?

Lines 167 and 174: space after 100.

Line 176 – spelling mistake

Line 237 (and throughout): SI (and other abbreviations) should be defined on first use.

Table 1: tables should be able to stand on their own.  Please explain more in the legend about how to interpret your results and add the DMSO control results to the table.

Figure 6: you’re a bit inconsistent with your labelling: only riboflavin has B2 following it, please also put the B numbers beside the others.   Figures should stand on their own, therefore please explain in the legend what each abbreviation stands for.  I think it would be confusing for people who are unfamiliar with vitamins that your figures are labelled with capital A, B, C, etc. because it looks as if those are the vitamins in the legend, therefore please change to 6.1, 6.2, 6.3 etc.   

Lines 322, 366 and elsewhere: no hyphen before the 2 in SARS-CoV2.

Line 352: don’t you mean to confirm the computer model predictions since plaque reduction is in vitro?

Line 367: explain that H-bonds are hydrogen bonds.

Line 386: colon after residues.

Line 439 and Figure 8: how do you know that binding to the spike is virucidal rather than neutralising?  I don’t understand Figure 8: if methylcobalamin was virucidal, wouldn’t inhibition be 100%?

Lines 448, 467, 470: references please.

Line 453: companioning isn’t a word.  Accompanying or compounding maybe?

Line 461: tended – past tense.

Line 463: therapeutic outcomes.

Line 461-464: oh tut tut – this is a very dishonest sentence – out of 64 studies of vitamin C and COVID you cite the ONLY TWO in which vitamin C did not do well!  Shame on you.

Line 474: reference 16 was a review, not a clinical trial. Make it clear that you are not referring to ventilators etc. in relation to COVID in this sentence.  Please check that your references are accurate throughout and not misleading.

Line 474-476: Again this is a shocking lie – reference 58 has absolutely NO MENTION of COVID or SARS-CoV2 and is a study which ended in 2015. 

I am not going to check all your references for you, I stopped reviewing the Discussion after this misrepresented reference: you have established a poor track record therefore you must check that ALL OF your references are accurate throughout and not misleading because if you re-submit and I find ONE SINGLE mistake of this sort I will reject your paper.

Line 540: this is the first mention of cobamamide!

Lines 603, 699, 701 and other references: too many periods. Please check all your references carefully for such mistakes.

Comments on the Quality of English Language

There are only a few minor problems with the English or grammar and I have noted most of them in my comments to authors.  On the whole, the paper is very well written.

Author Response

We appreciate the comments and suggestions of the reviewers. We have addressed each comment in detail and incorporated the changes within the revised manuscript. Please see the attached file for details.

Reviewer 2 Report

Comments and Suggestions for Authors

In their study, Moatasim et al. investigated the antiviral activity of a broad spectrum of different vitamins against viruses such as SARS-CoV-2, HCoV-229E, and MERS-CoV. The authors proposed a direct antiviral effect, particularly for vitamin B12, against the investigated viruses by interfering with receptor binding and viral replication in target cells. There is no doubt about the supporting role of certain vitamins in inducing immune responses, underscoring the importance of vitamin supplementation during viral infections. This current study reveals a direct antiviral effect of vitamins, particularly B12, in controlling SARS-CoV-2 and other coronavirus infections. The narrow non-toxic concentration range of vitamins showing some antiviral effect raises questions about how far this finding reflects the direct antiviral capacity of vitamins in vivo.

One of my major concerns is the use of CC50 values to calculate the safety index (SIs). The concentration of substances at CC50 already indicates substantial cytotoxicity and cell death. Therefore, SI values presented may overestimate the potential therapeutic window between antiviral effects and cytotoxicity. The highest effective concentration should be defined by the concentration of substances that do not cause any significant cytotoxicity. From this perspective, IC50 values, defined for example for B12 vitamin, indicate concentrations where cytotoxicity is already clearly visible.

Other concerns relate to the in vivo situation, particularly how relatively high concentrations can be achieved in vivo, especially on mucosal surfaces in the lungs, where viral replication is supposed to be inhibited.

Regarding the figures:

Generally, the labels on some figures (Figs. 2, 4, and 5) are too small and difficult or impossible to read. This issue should be addressed.

Figure 1: In case of some vitamins tested lower concentrations for viability and/or IC50 are not included. Why is that?

Figure 3: How many replicates/parallels are included? Is there information about standard deviation (SD)? Additionally, the color coding of viruses in Figure 2 and 3 should be matched.

Figure 4: The text in the figure is impossible to read.

Figure 5: The labeling is too small and difficult to read. The bars with color coding are also difficult to distinguish.

Figure 8: Are the differences between the different modes of action significant? If not, how can it be explained that B12 contributes equally through all investigated routes of action? Furthermore, the concentrations of Methylcobalamin used in this assay (over 10uM) seem to result in substantial cytotoxicity, as shown in Figure 2.

Author Response

(The authors gave the same response as above.)
